# Long-Term Pulmonary Function in Healthcare Workers: A Spirometric Evaluation Three Years Post-COVID-19 Pandemic

**DOI:** 10.3390/biomedicines13081809

**Published:** 2025-07-24

**Authors:** Lorenzo Ippoliti, Luca Coppeta, Giuseppe Bizzarro, Cristiana Ferrari, Andrea Mazza, Agostino Paolino, Claudia Salvi, Laura Angelini, Cristina Brugaletta, Matteo Pasanisi, Antonio Pietroiusti, Andrea Magrini

**Affiliations:** 1Department of Biomedicine and Prevention, University of Rome Tor Vergata, 00133 Rome, Italy; 2PhD Program in Social, Occupational and Medico-Legal Sciences, Department of Occupational Medicine, University of Rome Tor Vergata, 00133 Rome, Italy; 3Faculty of Medicine, Saint Camillus International University of Health Sciences, 00131 Rome, Italy

**Keywords:** COVID-19, spirometry, healthcare workers, pulmonary function, smoking, long-term effects

## Abstract

**Background**: The long-term impact of SARS-CoV-2 infection on pulmonary function remains insufficiently characterised, particularly among individuals who have experienced mild or asymptomatic disease. This study aimed to assess spirometric changes over a three-year period and evaluate potential associations with demographic and clinical variables. **Methods**: We retrospectively analysed spirometry data from 103 healthcare workers (HCWs) who underwent pulmonary function tests at three time points: before the pandemic (Time 0), one year post-pandemic (Time 1), and two years post-pandemic (Time 2). Linear regression models were employed to evaluate the impact of various factors, including age, BMI, gender, smoking status, history of SARS-CoV-2 infection, vaccination status prior to infection, and the number of infections, on changes in FVC and FEV_1_. **Results**: A statistically significant decrease in both FVC and FEV_1_ were observed at Time 1 and Time 2 compared to baseline (*p* < 0.05). Smoking habits were significantly associated with a greater decline in both FVC and FEV_1_. Multiple infections were associated with larger reductions in FVC at Time 1. No significant associations were found with age, gender, BMI, or vaccination status. Even in the absence of severe symptoms of the disease, healthcare workers exhibited a measurable decline in pulmonary function over time. Smoking and reinfection emerged as relevant factors associated with reduced lung capacity. **Conclusions**: These findings emphasise the need for ongoing respiratory monitoring in occupational settings and the importance of targeted preventive measures.

## 1. Introduction

The primary characteristics of SARS-CoV-2 are upper-respiratory and systemic symptoms, which are similar to those of influenza. Severe cases are characterised by lung damage, with fatal outcomes due to severe alveolar injury [1,2]. Unlike classic respiratory distress syndrome (ARDS), this damage is characterised by a mismatch between the severity of symptoms and hypoxaemia [3], which is also related with the occurrence of long COVID, a multisystemic condition comprising often-severe symptoms that follow a coronavirus 2 (SARS-CoV-2) infection [4]. The most common symptoms in patients that undergo this pathology are chest pain, fatigue, dyspnoea, and cough [5]. The exact causes are still under investigation, but could include organ damage, persistent inflammation or an immune response [6,7].

The COVID-19 pandemic has prompted extensive research into both the acute and long-term consequences of SARS-CoV-2 infection [7,8]. While much attention has been paid to severe cases requiring hospitalisation, there is increasing evidence that individuals with mild or moderate disease may also experience persistent respiratory symptoms and subtle impairments in lung function. These changes, although sometimes clinically silent, may have implications for long-term respiratory health and quality of life. Given the global scale of the pandemic and the large number of people affected, understanding the trajectory of lung recovery—particularly in those who are not hospitalised—is of considerable public health importance.

It is clear that COVID-19 produces an alteration in the lungs that is evident in patients with mild–severe infection [8], and also in children [9]. A systematic review found that around 40% of individuals who had recovered from COVID-19 showed impaired diffusing capacity of the lungs for carbon monoxide (DLCO), while 15% exhibited signs of restrictive lung disease, and 7% displayed obstructive patterns [10]. Indeed, DLCO was found to also be low in a considerable proportion of non-healthcare-seeking individuals 2 years after mild COVID-19 [11]. Pulmonary function tests have revealed lung alterations in patients diagnosed with COVID-19, even among those who appear otherwise healthy [12]. FVC (forced vital capacity) is a key parameter in the evaluation of COVID-19 patients, to assess both their initial condition and the recovery process [13].

However, studies have shown that patients recover lung function over time [14], but in some cases recovery even two years after the average infection may be limited [15]. Nevertheless COVID-19 survivors with mild/moderate disease usually did not suffer from long-term pulmonary sequalae and would not require routine follow-up [16].

In a preceding study, it was noted that FVC can decrease in patients who exhibit even mild respiratory symptoms [17,18,19]. The present study aims to investigate the progression of pulmonary function over time in a comparable population, three years after the onset of the pandemic. The primary objective of this study is to assess long-term changes in spirometric parameters among healthcare workers and to explore whether differences emerge based on SARS-CoV-2 infection history. The secondary objective is to explore any potential influence or association between SARS-CoV-2 vaccination and population characteristics, as highlighted in our earlier research [19].

## 2. Materials and Methods

In this study, we retrospectively analysed lung function tests from healthcare workers (HCWs) who underwent health surveillance visits at the Policlinic of Rome Tor Vergata between 2019 and 2024. At the policlinic, HCWs receive an annual spirometry test to assess their work fitness. Following the guidelines of the Italian Ministry of Health, routine pulmonary function tests were suspended during the SARS-CoV-2 pandemic (from March 2020 to September 2022) due to the elevated risk of COVID-19 transmission. To evaluate potential changes in respiratory function related to COVID-19 infection, we compared spirometry results from the six months prior to March 2020 (Time 0, T0) with those obtained from the same individuals during the year after September 2022 (Time 1, T1), and subsequently after September 2023 (Time 2, T2).

### 2.1. Data Collection

COVID-19 infection data were retrieved from the occupational health database. During the pandemic, all hospital HCWs were required to undergo a nasopharyngeal swab every 15 days, and the results were stored in a centralised database. Additionally, any symptomatic COVID-19 infections among these workers were routinely reported to the occupational health department and recorded in the same database [20]. As a result, data on both symptomatic and asymptomatic COVID-19 infections were available for all hospital employees. Information regarding COVID-19 vaccination was obtained from the regional vaccine database, which includes records of all residents who received the SARS-CoV-2 vaccine between 2020 and 2023.

### 2.2. Spirometry Test

Spirometric testing was conducted according to ERS/ATS standards [21]. All tests were performed using the same spirometer model (COSMED Quark PFT body with dongle, software OMNIA 1.5, Rome, Italy), calibrated daily, and operated by the same trained technicians throughout the study period. The three best measurements of FEV1 and FVC were selected.

### 2.3. Inclusion/Exclusion Criteria

Healthcare workers were eligible for inclusion in the study if they met the following criteria: having valid pulmonary function tests both before and after the pandemic (within the previously defined period); having no significant comorbidities, particularly respiratory diseases; the availability of molecular swab test results from the pandemic period; and the documentation of vaccination status. Subjects with severe COVID-19 infection, including those experiencing respiratory failure, septic shock, and/or multiple organ dysfunction or failure, were excluded from this study. As most infections were mild or asymptomatic and not clinically investigated with imaging, the presence of COVID-19-associated lung injury could not be determined for each participant.

### 2.4. Statistical Analysis

The following parameters were collected for each participant: gender, age, body mass index (BMI), smoking habits, history of infection with SARS-CoV-2 (number of episodes and dates), and vaccination status (number of doses). Spirometry test results from both the pre- and post-pandemic periods were recorded, including forced vital capacity (FVC) and forced expiratory volume in one second (FEV1). All spirometry data were documented as measured values (litres for volumes) and as percentages of predicted values. The data were entered into an Excel database for further analysis. Descriptive statistics were calculated for continuous variables, including mean, standard deviation, range, and percentage. Categorical variables were expressed as frequencies. Normality of distribution was verified using the Shapiro–Wilk test. Paired *t*-tests were used to compare spirometric values between T0 and T1, and between T0 and T2. A linear regression analysis was then performed to identify potential predictors of variation in spirometric outcomes. *p* values < 0.05 were considered statistically significant.

## 3. Results

A total of 103 subjects were included in the study, and their main characteristics are reported in Table 1. The mean age was 49.4 years (SD ± 9.8), and the mean BMI was 24.1 (SD ± 3.9). The sample was almost equally distributed by gender, with 50 males (48.5%) and 53 females (51.5%). Most participants were non-smokers (n = 84; 81.6%), while 19 individuals (18.4%) reported being smokers.

Regarding COVID-19 infection history, 63 participants (61.2%) had a confirmed SARS-CoV-2 infection, whereas 40 (38.8%) had no history of infection. Among those infected, the majority reported infection in 2022 (n = 51; 49.5%), followed by 2020 (n = 13; 12.6%) and 2021 (n = 9; 8.7%). Most of the infected individuals experienced a single episode of COVID-19 (n = 53; 51.5%), while 10 participants (9.7%) reported reinfection. Notably, 45 of the infected individuals (42.9%) had received the COVID-19 vaccination prior to their infection.

Table 2 summarises spirometric parameters (FVC and FEV_1_), showing the mean values and standard deviations at the three analysed time points. Paired *t*-tests were performed to assess differences in the mean values over time.

Table 3 and Table 4 present the findings of two linear regression analyses. The first model examines the relationship between SARS-CoV-2 infections in 2020, 2021, and 2022 and changes in FVC, while the second model examines the same relationships in relation to changes in FEV_1_.

Linear regression analyses were subsequently conducted to assess variation in FVC (ΔT1–T0 and ΔT2–T0) and FEV_1_ (ΔT1–T0 and ΔT2–T0). The independent variables considered were age, BMI, gender, smoking habits, history of SARS-CoV-2 infection, vaccination status prior to infection, and number of infections. The results of these analyses are presented in Table 5, Table 6, Table 7 and Table 8.

## 4. Discussion

This study aimed to evaluate the long-term evolution of pulmonary function among healthcare workers over three years following the peak of the pandemic, focusing particularly on changes in spirometric parameters such as FVC and FEV_1_ and their association with demographic and clinical variables, including smoking status, infection history, vaccination status, and reinfection. Our findings revealed statistically significant decreases in both FVC and FEV_1_ values one and two years after the pandemic’s peak, compared to pre-pandemic measurements. The trends observed in absolute values were confirmed by a review of percent predicted values, which also showed a consistent pattern of decline over time. This observation suggests that, even among healthcare workers who predominantly experienced mild or asymptomatic SARS-CoV-2 infections, subtle yet measurable respiratory functional declines may persist for years after initial exposure. However, while our regression analyses identified a significant association between the number of infections and FVC decline at one year (ΔT1–T0), no other infection-related variables, including infection per se, showed statistically significant associations. Therefore, although these results are consistent with the literature documenting post-acute sequelae of SARS-CoV-2 infection [17,18,19], the contribution of the infection itself to the observed decline remains uncertain. Unlike studies reporting moderate-to-severe impairment or persistent clinical symptoms, our data suggest a more gradual, subclinical alteration of respiratory physiology, potentially modulated by factors such as reinfection or smoking, which nonetheless merits attention given its long-term nature and potential implications for occupational health. Although the reduction in FEV_1_ was statistically significant, its magnitude (approximately 110 mL) fell within the range of normal intra-individual variability in healthy adults. Similarly, FVC is known to be influenced by patient effort and operator interpretation. Therefore, the clinical relevance of these findings should be interpreted with caution, even though population-level trends may still reflect subtle physiological changes.

Although BMI is known to influence pulmonary function, particularly through restrictive mechanisms, the relatively normal BMI range in our sample (mean 24.1) may have limited the ability to detect any such effect. The significant association between smoking and decreased spirometric parameters further corroborates the established role of tobacco exposure in impairing lung function and recovery [22]. Smokers in our cohort experienced a more pronounced decline in both FVC and FEV_1_, emphasising the combined effect of environmental factors and viral infection on respiratory health. Notably, the number of SARS-CoV-2 infections also emerged as a significant predictor of FVC variation at one year (ΔT1–T0), indicating a potential dose–response relationship whereby repeated viral infections may hinder the lung’s capacity for complete recovery or remodelling. However, this effect was not observed at the two-year mark (ΔT2–T0), which may indicate partial recovery over time or the influence of other unmeasured confounders, such as physical activity, occupational exposures, or pre-existing subclinical conditions. The absence of statistically significant associations between pulmonary function decline and demographic factors such as age, BMI, and gender suggests that these factors may play a limited role in predicting post-COVID-19 lung function trajectories within the relatively homogeneous and healthy working population studied.

Of particular note is the lack of a protective effect associated with prior SARS-CoV-2 vaccination. While vaccination has been shown to reduce the severity of acute illness and prevent hospitalisation and death, our data did not demonstrate a statistically significant association between prior vaccination and improved spirometric outcomes. This finding is consistent with previous research indicating that, while vaccines can mitigate the acute inflammatory and immune-mediated damage caused by the virus, they may not prevent the development of long-term respiratory complications in all cases. Similarly, the regression models failed to show significant associations between year of infection and changes in FEV_1_, although a significant impact on FVC was observed among those infected in 2021. This temporal association could reflect the influence of specific viral variants, such as Delta, which are known to have greater pulmonary tropism and pathogenicity than earlier or later strains. These results highlight the complexity of the long-term effects of SARS-CoV-2 and suggest that viral evolution, the host immune response, and broader environmental factors likely all contribute to the observed patterns of pulmonary recovery or decline.

When interpreting these findings, the strengths and limitations of the study must be considered. Key strengths include the longitudinal design, the use of objective spirometric measurements at three time points, and the comprehensive data on infection and vaccination status. Additionally, the exclusive inclusion of healthcare workers enabled certain occupational and surveillance biases to be controlled, given that these individuals routinely underwent testing and follow-up as part of institutional health protocols. However, the retrospective nature of the analysis and the lack of DLCO measurements or imaging data limit our ability to fully characterise the nature of the observed functional impairments. Furthermore, excluding individuals with severe forms of the disease may have resulted in an underestimation of the full range of long-term pulmonary consequences. Finally, while the study sample was relatively balanced in terms of gender and smoking status, the sample was restricted in terms of geography and profession, which may limit the generalisability of the findings to other populations. The absence of imaging or detailed clinical data prevents the stratification of participants by confirmed COVID-19-related lung injury. This limitation may influence the interpretation of the findings, as the presence or absence of structural lung involvement remains unknown.

In conclusion, this study provides further evidence that even mild or moderate cases of COVID-19 can lead to long-term reductions in pulmonary function, measurable years after the acute phase. Smoking and reinfection have emerged as relevant risk factors. While the overall impact on respiratory health appears modest and subclinical, these alterations may be particularly concerning in occupational settings or in individuals with pre-existing vulnerabilities, as respiratory capacity is fundamental to overall health and function. Further longitudinal studies incorporating a broader array of respiratory metrics, clinical assessments, and biomarkers are needed to clarify the underlying pathophysiological mechanisms and guide surveillance strategies and preventive measures for at-risk worker populations.

## 5. Conclusions

This study reveals a statistically significant but modest decrease in FVC and FEV_1_ among healthcare workers in the years following the onset of the SARS-CoV-2 pandemic, even among those who did not experience severe symptoms. These results imply that SARS-CoV-2 could induce subtle, enduring alterations to the respiratory physiology with important implications for occupational and public health. Greater functional decline was associated with smoking and reinfection, highlighting the need for targeted preventive strategies. Conversely, no significant associations were found with age, BMI, gender, or vaccination status, suggesting that the long-term impact may occur independently of these factors in relatively healthy working populations. These results support the integration of long-term lung function monitoring into occupational health protocols, and emphasise the need for further research into the mechanisms and clinical significance of this decline, as well as potential interventions, such as smoking cessation or pulmonary rehabilitation, that could aid recovery.

## Figures and Tables

**Table 1 biomedicines-13-01809-t001:** Characteristics of study population.

		n	%	Mean (S.D.)
Total		103		
Age				49.2 (9.8)
BMI				24.1 (3.9)
Gender	Male	50	48.5	
	Female	53	51.5	
Smoking habit	Non-smokers	84	81.6	
	Smokers	19	18.4	
COVID-19 infection	Negative	40	38.8	
	Positive	63	61.2	
Years of infection	2020	13	12.6	
	2021	9	8.7	
	2022	51	49.5	
Number of infections	1	53	51.5	
	2	10	9.7	
Vaccinated before infection		43	41.7	

S.D.: standard deviation.

**Table 2 biomedicines-13-01809-t002:** Main lung function parameters in the spirometry test.

Measured Parameter	Time of Analyses	Mean (S.D.)	CI (95%)	*p*
FVC (L)	T0	4.52 (1.02)		
	T1	4.41 (1.04)	0.64–0.16	<0.05
	T2	4.33 (1.05)	0.14–0.26	<0.05
FEV1 (L)	T0	3.60 (0.86)		
	T1	3.50 (0.85)	0.59–0.15	<0.05
	T2	3.49 (0.90)	0.06–0.15	<0.05

S.D.: standard deviation; CI: Confidence Interval.

**Table 3 biomedicines-13-01809-t003:** Impact of SARS-CoV-2 infection (2020–2022) on variation in FVC.

Year of Infection	B	S.E.	*p* Value	CI 95%
2020	1.6	1.9	Ns	−2.2–5.3
2021	4.9	2.2	<0.05	0.6–9.2
2022	0.3	1.3	Ns	−2.2–2.8

B: unstandardized regression coefficient; S.E.: Standard Error; CI: Confidence Interval; Ns: Not Significant.

**Table 4 biomedicines-13-01809-t004:** Impact of SARS-CoV-2 Infection (2020–2022) on variation in FEV_1_.

Year of Infection	B	S.E.	*p* Value	CI 95%
2020	3.2	2.0	Ns	−0.7–7.2
2021	4.1	2.3	Ns	−0.5–8.7
2022	1.2	1.3	Ns	−1.4–3.9

B: unstandardized regression coefficient; S.E.: Standard Error; CI: Confidence Interval; Ns: Not Significant.

**Table 5 biomedicines-13-01809-t005:** Linear regression: FVC (ΔT1–T0).

	B	S.E.	*p* Value	CI 95%
Age	0.8	0.7	Ns	−0.5–0.2
BMI	−0.2	0.2	Ns	−0.4–0.3
Gender	−1.3	1.3	Ns	−3.9–1.2
Smoking habit	2.2	0.8	<0.05	0.6–3.7
COVID-19 infection	−4.2	3.8	Ns	−11.7–3.2
Vaccination before infection	−0.6	1.9	Ns	−4.3–3.0
Number of infections	5.2	2.4	<0.05	0.49–9.9

B: unstandardized regression coefficient; S.E.: Standard Error; CI: Confidence Interval; Ns: Not Significant.

**Table 6 biomedicines-13-01809-t006:** Linear regression: FVC (ΔT2–T0).

	B	S.E.	*p* Value	CI 95%
Age	0.2	0.1	Ns	−0.1–0.4
BMI	0.0	0.3	Ns	−0.6–0.6
Gender	−2.9	2.2	Ns	−7.3–1.6
Smoking habit	3.9	1.4	<0.05	1.2–6.6
COVID-19 infection	−1.9	6.5	Ns	−14.8–11.0
Vaccination before infection	−3.7	2.2	Ns	−11.0–1.7
Number of infections	3.0	4.1	Ns	−5.2–11.3

B: unstandardized regression coefficient; S.E.: Standard Error; CI: Confidence Interval; Ns: Not Significant.

**Table 7 biomedicines-13-01809-t007:** Linear regression: FEV_1_ (ΔT1–T0).

	B	S.E.	*p* Value	CI 95%
Age	0.1	0.1	Ns	−0.9–0.2
BMI	−0.2	0.2	Ns	−0.5–0.2
Gender	−0.1	1.4	Ns	−2.9–2.7
Smoking habit	1.8	0.9	<0.05	0.1–3.5
COVID-19 infection	−2.6	4.1	Ns	−10.7–5.5
Vaccination before infection	−0.7	2.0	Ns	−4.7–3.3
Number of infections	4.7	2.6	Ns	−0.4–9.9

B: unstandardized regression coefficient; S.E.: Standard Error; CI: Confidence Interval; Ns: Not Significant.

**Table 8 biomedicines-13-01809-t008:** Linear regression: FEV_1_ (ΔT2–T0).

	B	S.E.	*p* Value	CI 95%
Age	−0.1	0.1	Ns	−0.2–0.1
BMI	0.1	0.2	Ns	−0.3–0.5
Gender	1.8	1.6	Ns	−1.3–4.9
Smoking habit	−1.9	0.9	<0.05	−3.8–−0.1
COVID-19 infection	−0.8	4.6	Ns	−9.9–8.2
Vaccination before infection	2.5	2.2	Ns	−1.9–6.9
Number of infections	−1.5	2.9	Ns	−7.3–4.2

B: unstandardized regression coefficient; S.E.: Standard Error; CI: Confidence Interval; Ns: Not Significant.

## Data Availability

Datasets used and/or analysed during the current study are available from the corresponding author on reasonable request.

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
