# Peer review of "Long-Term Pulmonary Function in Healthcare Workers: A Spirometric Evaluation Three Years Post-COVID-19 Pandemic"

_biomedicines, 2025, doi:10.3390/biomedicines13081809_

Round 1
Reviewer 1 Report
Comments and Suggestions for Authors
This paper is an interesting study on the long-term impact of COVID-19 infection. This manuscript is generally well-organized, and the author's efforts to address a measurable decline in pulmonary function over time in healthcare workers, in the absence of severe symptoms of the disease, are commendable. Although the study size is small, the data includes background information and pulmonary function test results, making it highly suitable for analysis and very meaningful. That said, I have several comments and suggestions that may help strengthen the paper's clarity, scientific rigor, and overall impact.
- In Tables 5, 6, 7, and 8, while the reduction in FVC from T0 to T1 is indeed associated not only with smoking but also with the number of infections as an independent factor, as you have stated, no other infection-related variables show significant differences. Therefore, the evidence is insufficient to conclude that the decline is attributable to COVID-19 infection.
- In this context, was there any difference in pulmonary function test results between infected and non-infected individuals?
- Additionally, the abbreviation "B" is used in the tables, but no corresponding footnote or explanation is provided.
Reviewer 2 Report
Comments and Suggestions for Authors
Dear authors
I read your manuscript with great interest. Below you will find some doubts and suggestions that I hope can contribute to the final version.
Line 60: The primary objective appears unclear — is it referring to individuals with similar characteristics who had a previous SARS-CoV-2 infection, or does it also include those without prior infection? Since COVID-19 is not listed among the inclusion criteria, and the sample includes non-infected individuals, it remains unclear to me what the actual focus of the study was intended to be. Is just to study lung function variation in healthcare workers?
Line 73: Reference is missing.
Line 100: Which spirometer was used? Was it always the same device? Because there may be some variation in results when using different flow sensors (and also operators, despite using the same guidelines)
Line 121: Although the sample size is 103, did you not check for normality?
Line 121: I believe this should be clarified: is it T0 with T1 and T1 with T2, or T0 with T2?
Line 182: Although it is statistically significant, it may not be clinically relevant, as the difference in FEV₁ between T0 and T2 is 110 mL, which falls within the range of intra-individual variability. As for FVC, it is a parameter that is highly dependent on patient cooperation and the operator's judgement in achieving a full inspiration and determining the end of the test, and should therefore be interpreted with caution.
Line 199: 4.52 – 4.41 = 0.11, which once again falls within the range of intra-individual variability. I understand that, when considering the means, a trend can be observed; however, I still believe that issues related to intra-individual variability — and even the technical aspects I mentioned earlier — should be addressed, as the variation is minimal.
Line 205: Given that a high BMI is known to potentially cause a restrictive pattern, particularly when fat distribution is central, I believe this result should be discussed in greater detail, taking into account the characteristics of the sample.
Line 240: I would recommend revisiting this statement, as respiratory capacity is a fundamental aspect regardless of the context
Line 246: Considering the previous remarks, I would suggest revising this sentence, particularly the use of the term 'notable'
Kind regards
Reviewer 3 Report
Comments and Suggestions for Authors
A review of “ Long-Term Pulmonary Function in Healthcare Workers: A Spirometric Evaluation Three Years Post-COVID-19 Pandemic” paper by L. Ippoliti, L. Coppeta, G. Bizzarro, C. Ferrari, A. Mazza, A. Paolino, C. Salvi, L. Angelini, C. Brugaletta, M. Pasanisi, A. Pietroiusti, and A. Magrini
The main question of the paper “Long-Term Pulmonary Function in Healthcare Workers: A Spirometric Evaluation Three Years Post-COVID-19 Pandemic” paper by L. Ippoliti, L. Coppeta, G. Bizzarro, C. Ferrari, A. Mazza, A. Paolino, C. Salvi, L. Angelini, C. Brugaletta, M. Pasanisi, A. Pietroiusti, and A. Magrini was to investigate the progression of pulmonary function over time in mild-to-moderate COVID-19 survivors three years after the pandemic.
The importance of this topic is due to a huge number of people whose lungs were injured during COVID-19 pandemic and scarce data published on the lung function recovery rate after COVID-associated lung disease.
The methodology of this study is appropriate to the study goals. The authors performed a retrospective analysis of 103 healthcare workers survived after both symptomatic and asymptomatic COVID-19 and underwent spirometric examination in 2020, 2022, and 2023.
The conclusions made by the authors are consistent with the results and they address the main question posed.
The references are appropriate to the topic and the results of the study.
The manuscript includes eight tables and no figures. The tables are consistent with the main task of the study. The data are shown in a clear manner.
My remarks are as follows:
- The healthcare workers included in the analysis experienced both symptomatic and asymptomatic COVID-19. As the lung function is the center of attention, it seems to be reasonable to indicate if all the patients had COVID-associated lung injury. If not, it is worth to select patients with and without the lung injury and to analyze the lung function separately in these groups.
- The authors presented FVC and FEV1 in absolute values only. It is well known that the lung function decreases with age, so the regression in the absolute values could be related to the ageing of the patients. The analysis of per cent of predicted values could demonstrate more clearly the changes in lung function associated with recovery from COVID-19 as the predicted values are adjusted to age.
The manuscript can be published after minor revision.
Round 2
Reviewer 1 Report
Comments and Suggestions for Authors
Thank you for responding to my comments. Your replies adequately addressed the issues I raised, and I have understood them. I have no additional comments.